# A Theoretical Thermal Tolerance Function for Ectothermic Animals and Its Implications for Identifying Thermal Vulnerability across Large Geographic Scales

**Agustín Camacho** [1,2,*] 🆔, **Michael J. Angilletta, Jr.** [2] **and Ofir Levy** [3]

1   Depto de Ecología Evolutiva, Estación Biológica de Doñana, Isla de la Cartuja, 41092 Sevilla, Spain
2   School of Life Sciences, Arizona State University, Tempe, AZ 85281, USA
3   School of Zoology, Tel-Aviv University, Tel Aviv 69978, Israel
*   Correspondence: agustin.camacho@ebd.csic.es or agus.camacho@gmail.com

**Abstract:** The time-to-thermal-death curve, or thermal death curve, seeks to represent all the combinations of exposure time and temperature that kill individuals of a species. We present a new theoretical function to describe that time in lizards based on traditional measures of thermal tolerance (i.e., preferred body temperatures, voluntary thermal maximum, and the critical thermal maximum). We evaluated the utility of this function in two ways. Firstly, we compared thermal death curves among four species of lizards for which enough data are available. Secondly, we compared the geography of predicted thermal vulnerability based on the thermal death curve. We found that the time to loss of function or death may evolve independently from the critical thermal limits. Moreover, the traditional parameters predicted fewer deleterious sites, systematically situated at lower latitudes and closer to large water bodies (lakes or the coast). Our results highlight the urgency of accurately characterizing thermal tolerance across species to reach a less biased perception of the geography of climatic vulnerability.

**Keywords:** thermal limits; critical thermal maximum; voluntary thermal maximum; time-to-death curve; preferred temperatures; climatic vulnerability

## 1. Introduction

Reliable and powerful predictions of a population's vulnerability to climate warming are paramount to guide the distribution of resources needed to preserve biodiversity [1]. While overly sensitive models will spill efforts over species and places that do not need it, insensitive models may fail to identify populations threatened by heat stress.

Populations can be considered thermally vulnerable when living at thermally deleterious sites. This is where environmental temperatures overcome individuals' thermal tolerance [2–6] in a way that impairs their performance, and either alone or in combination with negative interactions induces population decline. Under this definition, estimating a population's thermal vulnerability requires combining an estimate of its thermal tolerance with models of its exposure to stressful temperatures [2,3]. For a local population, the time and magnitude of heat stress (also called the pulse and press [4]) represent the duration of exposure to unsuitable temperatures and the amount of deviation from suitable temperatures, respectively. At geographic scales, the extent and distribution of vulnerable populations also become essential measures of thermal vulnerability for the whole species. Although the extent of vulnerable populations within a species' range will determine the level of climatic threat for its persistence, their distribution will help identify where to take measures and the organizations responsible for their conservation.

Several researchers have produced models of thermal vulnerability for species across the world, but most often based on single parameters of thermal tolerance e.g., [5–8]). For

example, studies of lizards have explored the use of different parameters for the distribution of preferred body temperatures (PBTs) [9–11] or their voluntary thermal maximum (VTmax) [12,13]. However, the critical maximum (CTmax), and more lately subcritical temperatures (SCTmax1h) or temperatures that kill under 1 h exposure, has been used in vulnerability studies of ectothermic animals [13–17]. Although analyses using preferred temperatures point to tropical terrestrial species as the most vulnerable [9,11], studies based on the CTmax [13,14] sometimes oppose this view, marking populations from subtropical and temperate latitudes as more vulnerable. This situation suggests that some of the parameters so far used for modelling climatic limits to distribution could be misrepresenting the geography of vulnerability. In parallel, approaches that include the whole range of species could be needed instead of the traditional approach of estimating vulnerability solely at the sites of tolerance measurement for each species [5,14].

The time of exposure to stressful temperatures is a key component of thermal tolerance that has been implicitly used in vulnerability studies. For example, many studies use the mean or the 75th percentile of a distribution of the PBT of the studied species, measured for active animals with a capacity for thermoregulation [9,11]. These studies typically assume that populations will become thermally vulnerable when they are subjected to prolonged periods (e.g., 2 months) of daily environmental temperatures reaching right over their parameter of PBT. When this happens, their activity and possibilities for energy gain become restricted. Instead, other studies using the CTmax or $S_{CTmax1h}$ assume that thermal risk derives from short exposures to temperatures over such temperatures [13,16].

Although both assumptions might be correct, there are important reasons to believe that different parameters could induce specific geographic biases when evaluating thermal vulnerability. For example, since long periods of steady warm weather are necessary to make populations vulnerable under the PBT approach, it might be most able to detect vulnerable populations at lower latitudes and coastal areas, where steady temperatures year-round are more likely. Instead, approaches based on an organism's CTmax might be most effective in more continental regions or latitudes at which higher heat peaks occur due to the combined effects of seasonality and continentality [17]. This situation poses two important consequences for the reliability of accepted global patterns in thermal vulnerability [9,13,14,18]. First, they might be heavily influenced by intrinsic geographic biases derived from using any single parameter. Second, estimates of thermal threats to biodiversity might be severely underestimated.

Instead, by including all of the possible deleterious time and magnitude combinations of thermal exposure to any given species, a thermal death curve (TDC) should reveal a more comprehensive description of heat stress and of the geography of thermal vulnerability. Notwithstanding, empirical curves created for fruit flies can predict heat damage in local populations [19]. However, creating such empirical curves typically requires identifying several time–temperature combinations that could lead to casualties among experimental individuals [20]. This process may take a long time to obtain the data and be dangerous for animals [21]. Thus, it would be most practical to use a single parameter to identify the most likely combination of exposure time and temperature magnitude that makes populations vulnerable to climatic extirpation. The voluntary thermal maximum (VTmax) exhibits qualities that might make it appropriate for such a function. This parameter lays at the upper end of the PBT distribution and represents a temperature perceived by the animal as so stressful as to avoid it even at the expense of facing potential predators [22,23]. Although the VTmax can be a few degrees lower than the CTmax [24], exposure to the former boosts energetic demands [25,26] and body water loss rates [27,28] with respect to preferred thermal levels and may induce function loss of death in less than 4 h (e.g., in lizards [23]). However, the performance of different thermal tolerance parameters to detect thermal vulnerability across geographic scaless has rarely been compared with a more comprehensive function describing thermal tolerance, such as the TDC.

In this report, we generate a theoretical TDC based on parameters of thermal tolerance estimated for several species of lizards (*Xantusia vigilis*, *Sceloporus occidentalis*, *Sphenomor-*

*phus quoyi*, and *Urosaurus ornatus*). Then, we generate predictions of thermally deleterious sites and evaluate the geographic vulnerability of one of these species (*U. ornatus*), for whom data on time-to-death [21,23,29] and body temperature values are available [30]. Our analysis focuses on three questions. First, does the TDC predict more thermally deleterious sites than a single parameter? Second, do the latitude, altitude, and continentality of deleterious sites, as well as the magnitude of predicted exposure to stressful temperatures, differ among thermal tolerance parameters? Third, how do maps of deleterious sites produced with different thermal tolerance parameters compare with the distribution of *U. ornatus*?

## 2. Materials and Methods

### 2.1. Generating a Thermal Death Curve

Following Arrhenius [31] and Rezende et al. [20], the TDC consists of an exponential decay function in which the survival time (measured in hours) depends on the temperature. We parameterized this curve as a function of an organism's CTmax, VTmax, and PBT:

$$\text{TDC} = \frac{-(CTmax - T) * DR}{(PBT - T) * (1 + exp(PBT - T) * DR)} \tag{1}$$

where *T* equals the organism's body temperature and *DR* equals the decay rate. With this equation, the decay in survival can be predicted from traditional parameters of thermal biology. When the body temperature equals the PBT, the TDC tends toward infinity, describing an asymptote at the PBT and no thermal constraints on life expectancy. When the body temperature reaches the CTmax, the function tends toward zero. The equation also describes an inflection point at the VTmax, making the time-to-death decay more slowly at temperatures over that thermal level.

The decay rate represents the thermal sensitivity of the time until death; that is, how much the life expectancy of individuals decays per increase in body temperature. This parameter can also be interpreted in the opposite direction, i.e., how much the time until death increases as the body temperatures decrease below the CTmax. The decay rate can be estimated by fitting the curve to data on observed times to death at different temperatures through non-linear least square models. We used the R base function "nls".

### 2.2. Exploring the Thermal Death Curve in Lizards

We fitted the thermal death curve to data on the time to death measured at different temperatures for four species of lizards (*Xantusia vigilis*, *Sceloporus occidentalis*, *Sphenomorphus quoyi*, and *Urosaurus ornatus*) representing three families (Xantusiidae, Scincidae, and Phrynosomatidae) [23,29,32–37]. To collect these data, researchers must kill animals, making such data rare in the literature. The decay rate was estimated for each species in which the PBT, VTmax, CTmax, and the time until death at two or more temperatures was known. We used a nonlinear least squares method to estimate the most likely decay rate given the data (see the supplementary R script). Figure 1 shows the observations and the fitted curves.

### 2.3. Modeling Thermally Deleterious Sites for Urosaurus ornatus

We modeled the geography of thermally deleterious sites for tree lizards (*U. ornatus*) for which enough data existed to parameterize the thermal death curve. We assumed that tree lizards are killed within one hour of exposure to their CTmax (the minimum temporal resolution at body temperatures were modeled). For the PBT, VTmax, and CTmax, we used the mean values of the PBT at 35.5° from Sinervo et al. [9] to make the conclusions more comparable, and of the VTmax at 42.52° and CTmax at 46.3 °C, which were obtained from the datasets "uro3" and "ctmaxuro" [23]. Other PBT estimates e.g., [23] differed by less than one degree from the one provided in Sinervo et al. [9]. We also obtained the time at which half of a sample of lizards lost their locomotor performance (4 h) from [23]. Lowe and Vance's data [29] enabled us to calculate a survival time of 1.27 h at a temperature of

44 °C, and Licht's data [21] yielded a survival time of 230 h at 41.5 °C (23 days of 10 h daily exposure, leading to three casualties with clear heat stress signs).

We also modeled the frequency and duration of body temperatures in tree lizards across North America. We developed an individual-based model of an adult lizard (snout vent length = 67 mm, mass = 10 g) based on a *Sceloporus* model [38], expanded by Levy and colleagues [30,39]. We used a published set of hourly microclimates [39] to calculate the operative temperatures of lizards on surfaces ranging from 0% to 100% shade. The operative temperature represents the steady-state temperature of an organism in a particular microclimate [40]. The microclimates spanned the United States and Mexico at a resolution of 36 km × 36 km for the period 1980–2000. For each hour, the air temperature, radiative load, and wind speed were used to calculate the lizard's operative temperature [40] in each microhabitat. These operative temperatures were stored as netCDF files, which can be processed by the R package "ncdf4" [41] as NC objects, dedicated to store data in an explicit spatiotemporal framework. By solving heat exchange equations [42], we calculated these body temperatures as:

$$T_{b,t} = T_{b,t-1} + \Delta T_b \qquad (2)$$

by solving previous heat exchange equations [42], where *t* is the current time, $T_{b,t}$ is the body temperature at the time *t*, $T_{b,t-1}$ is the previous body temperature, and $\Delta T_b$ is the change in body temperature after a period of time, $\Delta t$, which enhanced the stability of the model. In the model, the lizard can be active whenever it can attain a body temperature between 30.7 °C and 37.5 °C (central 75% of field body temperatures [23]). For simplicity, we assumed that a lizard maintains around the mean of PBTs when active (35.9 °C [23]) by shuttling between microclimates. If the body temperature was above the activity range, we assumed that the lizards could escape high temperatures by burrowing into the shade 10 cm below ground. At night, the lizards were assumed to be resting in fully covered microhabitats.

To detect time–temperature combinations that should induce population damage, we extracted from the netCDF files the following information: (1) the mean number of days where the body temperature exceeded the CTmax (days/year); (2) the mean number of days where the body temperature was above the VTmax for at least four hours (days/year); (3) the percentage of years where the body temperature was above the preferred body temperature (PBT) for at least four hours over more than 60 days (index used in [9]); (4) the number of hours where the body temperature exceeded a lethal combination of temperature and exposure time, as determined in the TDC (Equation (1)), at a resolution of 0.2° incremental steps from the PBT to the CTmax (hours/year). All of these were yearly variables that were averaged across the 20-year period of the database.

### 2.4. Geographic Bias in Predictions of Thermally Deleterious Sites for Urosaurus ornatus

We tested for biases in the geography of predicted thermally deleterious sites stemming from the chosen parameter of thermal tolerance. Specifically, we compared the latitude, altitude, and distance to the coast of thermally deleterious areas, generated with the PBT, VTmax, CTmax, or the TDC. We performed a generalized least squares test (using the R package "nlme" [43]) to compare the mean latitude, altitude, and distance to the coast for the predicted deleterious sites obtained with each thermal tolerance measure.

### 2.5. Comparison of Thermal Vulnerability Layers with Known Locations of U. ornatus

To explore how the maps of predicted deleterious sites might restrict the geographic distribution of the species, we overlapped known locations for tree lizards with the four raster layers of thermal vulnerability (Figure 4). Then, we calculated how many locations were listed as vulnerable by each raster. To do this, we downloaded all records available for *U. ornatus* in the Global Biodiversity Information Facility (GBIF) in March 2023, on the basis of "human observations". These records were checked to remove double zeroes, outliers, mirrored records, and records in the sea.

## 3. Results

### 3.1. Exploring the TDC in Four Lizard Species

The fitted curves and published data show that species differ not only in their values of CTmax but also in their decay rates. The CTmax values of the species were 46.3 °C for *U. ornatus* [23], 41.5 °C for *X. vigilis* [36], 44.5 °C for *S. occidentalis* [34], and 40.8 °C for *S. quoyi* [32]. The decay rates were 4.14 for *U. ornatus*, 135.6 for *X. vigilis*, 0.95 for *S. occidentalis*, and 2.05 for *S. quoyi*. Figure 1 suggests that a species with a higher CTmax (e.g., *S. occidentalis*) may still tolerate a subcritical temperature for less time than a species with a lower CTmax (e.g., *X. vigilis*).

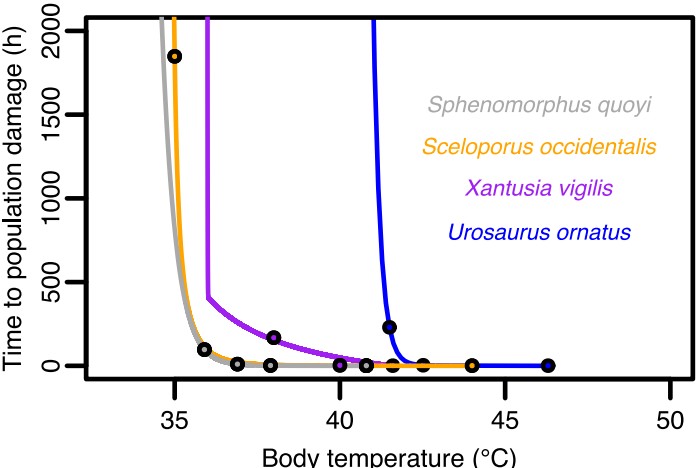

**Figure 1.** The expected relationships between the body temperature and the time until death varied among four species of lizards. These thermal death curves were fit to survival times at specific temperatures taken from the literature.

### 3.2. Geographic Bias in Predictions of Thermally Deleterious Sites for Urosaurus ornatus

The geographic distribution of thermally deleterious sites depended on which measure of thermal tolerance was used in the model. The predicted regions based on the CTmax or PBT were much smaller and spatially segregated than those predicted by the VTmax or the TDC (Figure 2). In fact, the regions predicted from a model based on the CTmax or PBT were often a subset of the regions predicted from a model based on the VTmax or TDC. Maps based on the TDC predicted a higher frequency of thermal impacts at higher latitudes compared to other descriptors of thermal tolerance. With respect to the altitude, the TDC predicted more deleterious sites at higher altitudes than the VTmax and the CTmax but comparable to the PBT (Figure 3, Table 1). Regarding the distance to the coast, the TDC predicted sites further from the coast than the PBT but closer than the VTmax and similar to the CTmax. In general, the VTmax and the TDC predicted more extensive climatically deleterious areas than the CTmax and PBT (Figure 3, Table 1).

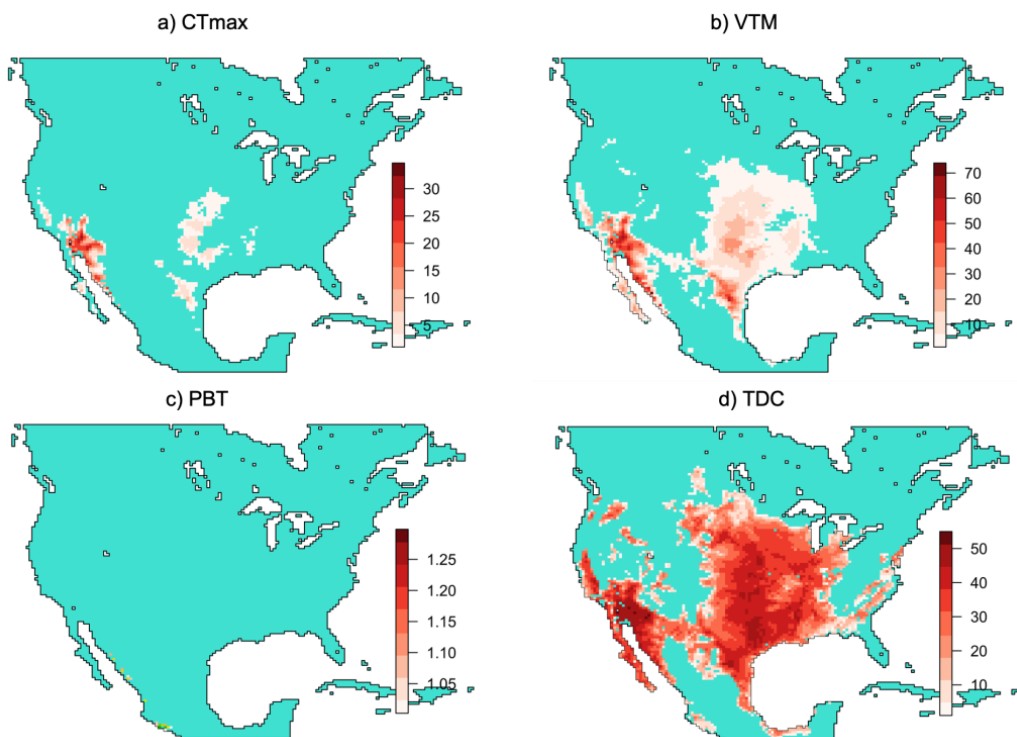

**Figure 2.** Maps of thermally deleterious sites for *Urosaurus ornatus* across North America between the years 1980 and 2000. Each map was derived from a model of thermal tolerance using one of four measures: the critical thermal maximum (CTmax), voluntary thermal maximum (VTmax), preferred body temperature (PBT), and thermal death curve (TDC). For each map, the colored legend represents the number of thermally deleterious events that would have occurred, defined as thermal stress that would kill at least half of the population. Only one hour of exposure to the CTmax would lead to a deleterious event. Lethal events for the model based on the VTmax and the model based on the PBT were 3 h and 60 days of exposure, respectively; see the Methods for details. Finally, the TDC represents all combinations of body temperature and the time of exposure required to kill a lizard.

**Table 1.** Results of a generalized linear model comparing how the latitude, altitude, and distance to the coast for predicted thermally deleterious sites vary with the measure of thermal tolerance. The parameters of the model show how predictions based on the preferred body temperature (PBT), the voluntary thermal maximum (VTmax), and the critical thermal maximum (CTmax) compare to predictions made under the TDC. Zero means lower than 0.001.

| Parameter | Value | St.Err. | *t*-Value | *p*-Value | Geographic Bias |
|---|---|---|---|---|---|
| (Intercept) | 34.059 | 0.104 | 327.383 | 0 | Latitudinal |
| PBT | −16.497 | 1.592 | −10.335 | 0 | Latitudinal |
| VTmax | −2.004 | 0.173 | −11.545 | 0 | Latitudinal |
| CTmax | −3.164 | 0.285 | −11.09 | 0 | Latitudinal |
| | | | | | |
| (Intercept) | 550.200 | 6.583 | 83.583 | 0 | Altitudinal |
| PBT | −68.824 | 100.995 | −0.681 | 0.495 | Altitudinal |
| VTmax | −67.636 | 10.981 | −6.158 | 0 | Altitudinal |
| CTmax | −154.647 | 18.052 | −8.566 | 0 | Altitudinal |
| | | | | | |
| (Intercept) | 480,059.4 | 5014.67 | 95.73100 | 0 | Water distance |
| PBT | −407,682.8 | 76,937.02 | −5.29892 | 0 | Water distance |
| VTmax | 60,374.2 | 8365.90 | 7.21670 | 0 | Water distance |
| CTmax | 23,370.1 | 13,752.33 | 1.69936 | 0.089 | Water distance |

### 3.3. Relating Predicted Thermally Deleterious Sites with Urosaurus ornatus Locations

Among the 2347 retrieved locations where *U. ornatus* has been observed, the number of locations predicted to be at thermally deleterious sites (i.e., thermally vulnerable) was

highest for the map based on the TDC (568), followed by the map based on the VTmax (170). The maps based on the CTmax or PBT did not predict any of the observed locations to be thermally vulnerable.

## 4. Discussion

We have provided a thermal death curve [20,31] mostly based on traditional thermal tolerance parameters. Our equation was motivated by previous studies [43–45] showing that one can predict the CTmax from survival times at lower temperatures. Until now, TDCs have been obtained by measuring the time until mortality over a range of high temperatures (reviewed in [20]). However, the shape of this curve remains unknown for lower temperatures.

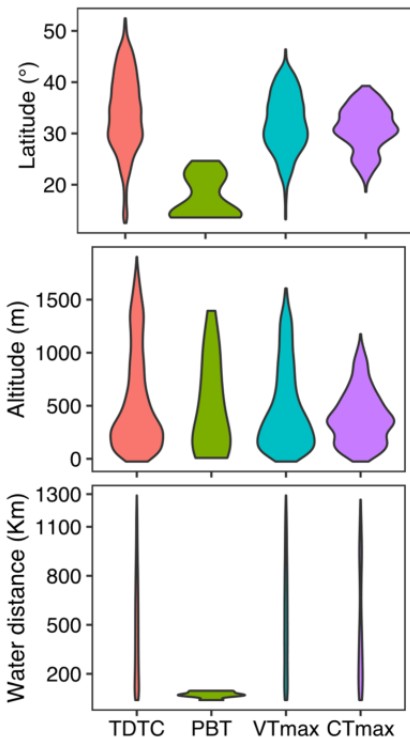

**Figure 3.** The frequency of thermally deleterious sites predicted for *Urosaurus ornatus* varied with the latitude, altitude, and continentality, represented here by the distance from major water bodies with respect to the measure of thermal tolerance used to predict deleterious sites: the thermal death curve (TDC), preferred body temperature (PBT), voluntary thermal maximum (VTmax), and critical thermal maximum (CTmax). The width of each violin plot represents the frequency of each value of the response variable, estimated as the kernel density.

The thermal death curve enables one to model the time until death over a wide range of temperatures, from the preferred body temperature to the CTmax. Although we assumed an infinite time of death at the preferred temperature, one could add a parameter for longevity, representing the maximum time until death for the organism. Doing this might be useful to compare species with very different life expectancies, but the increase in the complexity of the function and associated calculations prevented us exploring that parameter in this study. Moreover, we focused on creating a TDC to identify restrictions on survival at stressful temperatures rather than life expectancy at benign temperatures, which likely depends on many other factors.

The VTmax creates an inflection point in the thermal death curve, shifting it from convex to concave (i.e., slowing the decay rate at temperatures over the VTmax). This is supported by studies of *U. ornatus* showing that one and a half degree below the VT-max (41 °C), the time-to-death is relatively long (around 230 h) [21], while if the VTmax

(42.52 °C) lays below 4 h and at the CTmax (46.3 °C) this may happen under 1 h [23]. This deceleration could be caused by the induction of protective physiological or biochemical processes triggered when body temperature rises. Still, studies need to ascertain whether the VTmax acts as a trigger for these processes or if they onset at lower (more likely) or higher temperatures. Our TDC provides a formal way to explore relationships between classic thermal tolerance parameters, the temporal dimension of heat tolerance, and the subsequent geographic distribution of thermally deleterious sites. Jorgensen et al. [44] introduced a coefficient of thermal sensitivity, z, by fitting a linearized model of an empirical TDC that directly relates to our decay rate parameter. Our exploratory comparison of curves among four species and the above-cited data [21,23] suggest that the decay rate might evolve independently of the tolerated temperature range. If so, species with similar values of CTmax might differ in the time they tolerate subcritical temperatures, as seen for *X. vigilis* and *S. occidentalis*. The thermal sensitivity constitute a key factor for understanding how the thermal tolerance of organisms translates into geographic patterns of deleterious sites and populations' climatic vulnerability. Decay rates (DR) represent it in our function, making the TDC steeper as DR increases (Supplementary Figure S2). The implications of its variation require further exploration.

The thermal death curve can also be used to study the benefits of thermal acclimation. Adaptive thermal acclimation increases the time until death at high temperatures. As we saw in our geographic models, a large portion of deleterious sites may arise from exposure to subcritical temperatures for long periods (e.g., between 43 and the CTmax in our study, Supplementary Figure S1). Willot et al. [45] did not find significant changes in the times until death among six species of ants acclimated to 17, 26, and 30 °C. However, these northern temperate species might be somehow less selected to resist subcritical temperatures than southern species (see the increased times to death for the more southern species, *Lasius emarginatus*, in Figure 4 of their manuscript). Instead, as found for fruit flies, acclimation may increase the time to death during acute exposure at high temperatures while decreasing this time during chronic exposure at lower temperatures [46]. More studies are needed to understand the consequences of such changes on the geography of thermal vulnerability.

### 4.1. Predicting Population Vulnerability to Combinations of Temperature and Exposure Time

The physiological processes and temporal scales at which thermal stress causes population decline vary along the range of tolerated temperatures. Concerning the processes, exposure to environmental temperatures right above the PBT is not expected to kill individuals immediately. In turn, it might limit their foraging times and impair their reproductive rates [9]. Additionally, the rates of energy expenditure and water loss increase exponentially with temperature [27,28], limiting an organism's capacity to store nutrients. At temperatures near the VTmax and CTmax, the aforementioned processes occur exponentially faster (e.g., for the metabolic rate and water loss [26]), leading to death within a few hours [23,24].

Thus, organisms should seek thermal refuges within hot landscapes [47,48] by avoiding direct solar radiation and hot surfaces. In this way, at sites where local minimal operative temperatures reach a species' VTmax, its local population should be vulnerable to thermal stress. Even if the local thermal heterogeneity is greater than predicted by geographic models of environmental temperatures [49], individuals should accumulate within thermal refuges, leading to potential death if trapped in less buffering microhabitats (e.g., under a hot rock [50]). This situation jeopardizes populations in many ways, including via resource depletion within refuges, excessive risk of predation, exposure to dehydration, or heat shock.

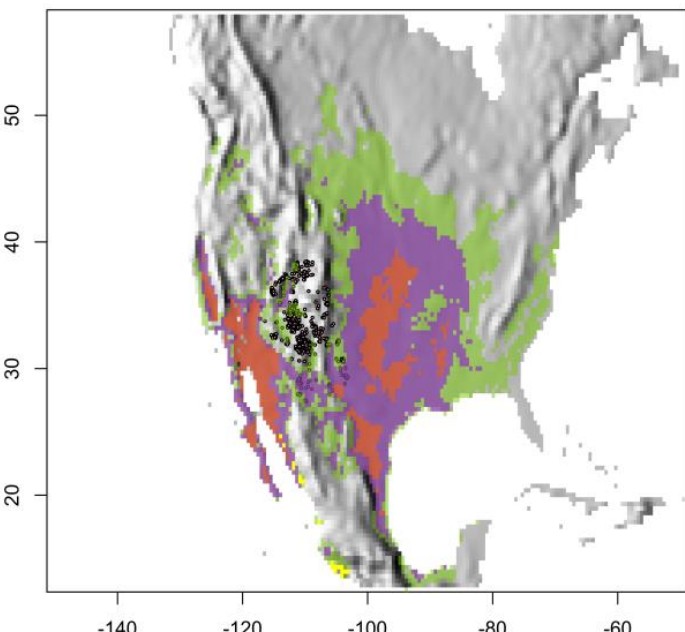

**Figure 4.** The maps of predicted deleterious sites shown in Figure 2 overlapped with locations (dots) at which *Urosaurus ornatus* has been observed. The map based on a thermal death curve (green) overlaps with 796 sites, while the map based on the voluntary thermal maximum (purple) overlaps with only 170 sites. Neither the map based on the critical thermal maximum (orange) nor the one based on the preferred body temperature (yellow) overlap with any location.

The minimum available temperatures in the hottest period of the year could be estimated using bioclimatic variables. For instance, the bioclimatic variable Bio5 represents the monthly maxima in shaded air temperatures, averaged across the warmest months of the past 10–20 years [51]. Under the assumption that this estimate reasonably represents the minimum available temperatures for a given species at any given site, it would flag deleterious sites whenever it reaches the species' VTmax at sites with known populations. At least these populations would had undergone consecutive situations of thermal vulnerability across the last decade and likely experienced thermally induced population declines. Within this context, both the VTmax and other sub-critical temperatures that impair sperm development have been used to successfully predict the thermally deleterious sites and geographic limits of lizards and other ectotherms [12,51,52]. Nonetheless, interpreting the effects of exposure to the CTmax is much more straightforward; population losses should always be expected whenever a reliable estimate of the minimum temperature available for a population of organisms reaches its average CTmax, or even better, if the data are available, the median Ctmax, which could be interpreted as an LD50 for the population. Finally, when the TDC is used to identify vulnerable sites, all of these processes discussed above contribute to identifying a site as thermally deleterious. This is why the model based on the TDC predicted the largest number of deleterious sites.

Further empirical and theoretical improvements should enable researchers to better combine thermal environments with a thermal death curve to identify vulnerable populations. Our TDC only identified sites as deleterious if the time until death or loss of function was achieved within a single period of continuous exposure to a given temperature. However, studies have shown that heat injury can accumulate across intermittent exposures to subcritical temperatures [21,44]. Thus, it remains to be better understood how thermal injury accumulates across days, including across recovery times, and how the accumulation of stress relates to the geography of vulnerable populations.

### 4.2. The Thermal Death Curve and Geographic Biases in Predicted Thermally Deleterious Sites

Taking the TDC as a more comprehensive measure of thermal tolerance and of geographic patterns of thermal vulnerability, we found that using different thermal tolerance parameters may bias the predicted geography of thermally deleterious sites. The systematic geographic biases found raise doubts about previous interspecific global patterns in thermal vulnerability. In these studies, each species' thermal vulnerability was estimated from a single parameter of thermal tolerance at a single location [5,6,9,13,14,18]. Using a single parameter of thermal tolerance at a single location per species likely gives an inaccurate picture of the thermal vulnerability or its geographic distribution. Additionally, geographic variation in thermal vulnerability can also derive from local processes such as acclimation or adaptation, which decrease vulnerability [53–55], or from dehydration, which increases it [51]. Since we were solely interested in exposing geographic biases derived from using different thermal tolerance parameters, we did not account for the effects of these other factors here.

### 4.3. Geographic Patterns in the Pulse and Press of Heat Stress

Mediated by thermoregulatory behavior (i.e., the VTMax), the thermal tolerance interacts with the time of exposure (known as the pulse) and level (known as the press [4]) of the thermal stress, shaping the geography of thermal vulnerability in each species. The long chronic exposures to temperatures right above the PBT that are needed to impair reproduction (e.g., 2 months, [9]) require steady exposures to benign temperatures. In our analysis, they were predicted at lower latitudes and closer to major water bodies. In contrast, the CTmax detected deleterious events at sites where continentality and seasonality amount to higher heat spikes [56]. In this study, we focused on deleterious combinations of the exposure time and thermal level, which we have referred to as deleterious events. However, to find regions where natural selection would occur, either based on thermal levels and on the time until loss of function, future studies could use TDCs to analyze the geography of the press as the locally induced magnitude of deviations from the TDC and of the pulse as the time of temperatures over deleterious levels separately. Combined with such studies, evaluations of the state of populations will help verify the predictive ability of different models of thermal vulnerability.

The power to predict thermally deleterious sites may also vary strongly with the parameter of thermal tolerance used, being maximal when using the TDC and followed by temperatures between the VTmax and CTmax (supporting online Figure S1). For the case of the tree lizard across North America, our constructed TDC found many more sites than any other combination, despite it not finding any thermally deleterious sites for temperatures below the VTmax (42.52 °C). The PBT detected very few deleterious sites, even though we relaxed Sinervo et al.'s [9] assumption of consecutive days with temperatures over the PBT in order to find more thermally deleterious sites.

### 4.4. Relating Predicted Thermally Deleterious Sites with Urosaurus ornatus Locations

The numbers of locations at which *U. ornatus* was predicted to be thermally vulnerable varied in a similar fashion across thermal tolerance measures. Maps based on the TDC and VTmax predicted many more thermally vulnerable populations than maps based on the CTmax or PBT. Under the assumption that all these locations correspond to still-existing populations, we would suggest that the TDTC and VTmax can be considered as earlier identifiers of thermally vulnerable populations than the CTmax and PBT. However, studies integrating field sampling, taxonomic validation, and thermal tolerance assays [57] are needed to assess the current status of these populations. In our example, we used automatically cleaned locations from the GBIF database of human observations, which include specimens deposited in scientific collections and observations uploaded to citizen science platforms such as I-naturalist. Thus, we cannot guarantee that all known locations were considered here. Our maps of thermally deleterious sites surround the known distribution of *U. ornatus,* potentially threatening the establishment of this species in these

sites, at least where deleterious events happen often or are very intense. Additionally, the TDC map led to a great extension of predicted thermally deleterious sites towards the north, which might be naively considered safe. This suggests that maps of thermally deleterious conditions could guide efforts in translocating thermally vulnerable populations [58].

Given our results, we cannot recommend using the PBT to identify thermal vulnerability. Researchers have used the mean and 75th percentile of the PBT distribution to predict the thermal vulnerability of populations e.g., [9–11]. These studies compared these two thermal tolerance measures with the mean of environmental temperatures instead of the minimum temperature available as we did. Although the first option greatly increases the likelihood of qualifying a population as vulnerable, choosing the latter descriptor of the environmental temperature is essential because in hot environments an ectotherm's options for thermoregulation can actually increase with increasing mean temperatures but always decreases when the minimum temperature increases [59].

Calculating all parameters of the thermal death curve (the time to function loss at different temperatures, PBT, VTmax, and CTmax of a species) becomes necessary for comprehensive predictions of vulnerability to heat stress and selective pressure on the time until loss of function. Still, experiments designed to measure the time until death at several temperatures raise ethical concerns, particularly for studies of vertebrates. Therefore, such data will be hard to accumulate for a large sample of species. Thus, we envision at least one way to lessen animal suffering while improving the detection of thermally deleterious sites for animals, namely to identify sites of thermal vulnerability as places where the minimum operative temperature reaches the VTmax, without regarding the time needed to cause death at this temperature. Measuring the VTmax is much faster than the PBT and safer than the CTmax [23], and the lower power of this trait to detect northern sites (compared to the TDC) could be palliated by reducing the time of exposure necessary to count a site as deleterious (see Supplementary Figure S2).

## 5. Conclusions

We still have a long road ahead to understand our tools to identify climatic vulnerability across geographic scales, so we should be cautious using published patterns of thermal vulnerability. Since actions mitigating the effects of climate warming are very costly, the identification of vulnerable populations should align well-identified records of geographic distribution with adequate thermal tolerance measures to map the thermal vulnerability of species across their geographic range and thermally deleterious sites around it.

**Supplementary Materials:** The following supporting information can be downloaded at: https://www.mdpi.com/article/10.3390/d15050680/s1. Figure S1: Number of vulnerable sites detected using different temperatures and their respective times needed to kill individuals, as predicted by a TDTC constructed for Urosaurus ornatus. Figure S2: effects of changing the DR value on the shape of a thermal death curve with the parameters of *U. ornatus*. The higher it is, the steeper the curve becomes. Figure S3: Map of vulnerable sites generated by the VTM (left) with no time to function loss, compared to predicted sites by the VTM with 3 h exposures (right), as plotted in Figure 2. Table S1: Lizard parameters used to calculate changes in body temperature (ΔTb) of an *S. undulatus* lizard.

**Author Contributions:** Conceptualization, A.C.; methodology, O.L., A.C. and M.J.A.J.; software, O.L.; validation, O.L.; formal analysis, A.C.; investigation, A.C.; resources, A.C. and O.L.; data curation, O.L. and A.C.; writing—original draft preparation, A.C.; writing—review and editing, O.L. and M.J.A.J.; visualization, A.C. and O.L.; supervision, M.J.A.J.; project administration, A.C.; funding acquisition, A.C. All authors have read and agreed to the published version of the manuscript.

**Funding:** This study was funded with an MSCA-IF fellowship (897901) and a FAPESP-BEPE grant (19/07090–1).

**Institutional Review Board Statement:** "Not applicable". This study was based on already published data only.

**Data Availability Statement:** All the data and scripts to reproduce the analyses are available in the supporting information file. The scripts to generate the microclimatic models belong to O.L.

**Conflicts of Interest:** The authors declare no conflict of interest. The funders had no role in the design of the study; in the collection, analyses, or interpretation of data; in the writing of the manuscript; or in the decision to publish the results.

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
