# Peer review of "A Theoretical Thermal Tolerance Function for Ectothermic Animals and Its Implications for Identifying Thermal Vulnerability across Large Geographic Scales"

_diversity, doi:10.3390/d15050680_

Round 1
Reviewer 1 Report
The authors present a theoretical thermal tolerance function for ectothermic animals for estimating the thermal vulnerability across geographic regions. This is a challenging task and urgently needed in the context of climate warming. Although they constructed a legit mathematical approach for the biophysical process to address this question, my major concern here is the predictive power of the TDC model since reproductive success rather than individual thermal tolerance could be more closely related to population dynamics. It could be more persuasive if historical population data for the four model species in this study confirmed the conclusions of this theoretical model. Although there are limitations, I acknowledge that this analysis pipeline as a methodological paper would attract wide attention among entomologists and herpetologists.
Major comments
I'm curious if it's possible to model thermally harmful sites for at least two species (with differing thermal tolerance). This allows us to see how large biological traits influence thermal vulnerability across large geographic scales.
Minor comments
1. L177 climatic NC is a format of raster or any other abbreviation, could be more specific
2. L309 “TTDC”?
Author Response
Response to reviewer 1
The authors present a theoretical thermal tolerance function for ectothermic animals for estimating the thermal vulnerability across geographic regions. This is a challenging task and urgently needed in the context of climate warming. Although they constructed a legit mathematical approach for the biophysical process to address this question, my primary concern here is the predictive power of the TDC model since reproductive success rather than individual thermal tolerance could be more closely related to population dynamics. It could be more persuasive if historical population data for the four model species in this study confirmed the conclusions of this theoretical model. Although there are limitations, I acknowledge that this analysis pipeline as a methodological paper would attract wide attention among entomologists and herpetologists.
Dear reviewer, in this new version of the article, we provide a simple evaluation of the ability of each of the thermal vulnerability layers to flag vulnerable sites for Urosaurus ornatus. We also added more analyses (an exploratory analysis of variation in DR in the supporting file and altitudinal biases in predicting thermally deleterious sites).
Major comments
Is it possible to model thermally harmful sites for at least two species (with differing thermal tolerance)? This allows us to see how large biological traits influence thermal vulnerability across large geographic scales.
Although we now provided a comparison of the layers with Urosaurus locations, making maps for more species would make the article overlong. Still, one next article is underway to explore sources of variation in predictions made with the TDC.
Also, please consider that this article only wishes to show that different metrics of thermal tolerance lead to very different geographies of expected thermal vulnerability. How much temperatures constrain the distribution of each species is a bit different, although of course related, problem.
Minor comments
- L177 climatic NC is a format of raster or any other abbreviation, could be more specific
We added: “These operative temperatures were stored as netCDF files, which can be processed by the R package “ncdf4” [57] as NC objects, dedicated to store data in an explicit spatio-temporal framework”.
- L309 “TTDC”?
TDTC thermal death time curve, thanks for spotting that, which is now corrected.
Reviewer 2 Report
see attached pdf.

Author Response
Response to reviewer 2
In this manuscript, the authors describe a new method for estimating the vulnerability of ectotherms to climate warming. By using non-lethal parameters to create a thermal death curve, the authors predict the amount of time at a given temperature it takes for a population to experience lethal effects of temperature. They then compare results of this curve to results from using individual parameters like the CTmax and voluntary thermal max. I commend the authors for undertaking such a task as this will be useful to a diversity of researchers considering effects of temperature on wild populations. I have personally conducted numerous experiments on reptile embryos which required heating eggs to the point of death (much to my dismay). This paper has actually allowed me to consider some ways of incorporating the present study into my own research program. I am very grateful. I think the manuscript is mostly well-written, particularly the discussion is well-done and covers all the necessary discussion points effectively. I do have a few major concerns with the manuscript, however. In particular, there are some discrepancies between the equation presented in the text and what is used in the R code that need attention. I also think the authors could do some additional work to validate the use of the TDC. I outline these primary concerns below followed by some trivial concerns. I am confident that the authors can address my concerns with a major revision of the manuscript, and I would be willing to re-evaluate the paper once these concerns are addressed.
Dear reviewer, thank you for your attentive review of our manuscript. The development of this curve has been complex and very long, with many previous steps and versions. We apologize for mixing parts from different versions and caused by sending the manuscript in a hurry due to a tenure contest. Hopefully, there is no confusion in this new version. Since the corrections altered a lot the number of lines and we also had to move around some paragraphs to insert the figures better, instead of text lines, we included in our responses the actual parts of the modified manuscript text. Besides we also edited the text for the correctness and to present a more matured flow of ideas.
Major concerns
I have several concerns regarding the generation of the Thermal Death Curve.
- The equation provided in the manuscript (Equation 1) does not match the equations used in the R code. Each R script provided, in the numerator of the latter portion of the equation, reads “CTmax - T” but the equation in the text reads “CTmax - PBT”. The authors needs to clarify if this variable should be “body temperature” (T) or “preferred body temperature” (PBT).
The function that models the curve in the script (filename: Curve DR and figure) is:
Which translates into the R function:
cu=function(t){
((-(c-t)*DR/(a-t)*(1+exp((b-t)*DR))))
}
Where t is body temperature, a is PBT, b is the VTM and c is the CTMAX
We added that clarification to the script and changed the original I value by DR, as used in the manuscript.
- The equation given in the text includes a longevity variable (L); however the equations in the R code do not include this variable. There are a few problems with this.
- a) Foremost, although the authors indicate that omitting Longevity has no major
impact on the curve itself, this appears not to be the case. I ran several iterations
of their equation including multiple values of Longevity (from 1, 5, 10, and 20 year
lifespan) and this altered the shape and location of the TDC dramatically. I think
some clarification is needed.
We apologize for this remnant of a previous manuscript version. Including longevity is difficult because it actually alters the final definition of the curve and its capacity to adjust to observed data. Besides, maximum longevity depends heavily on other factors apart from temperature. Thus, we decided to remove the longevity term from the final equation (the one used to make the graphs), although this passed unnoticed in the text. Thus, equation 1 is actually no longer correct, and we changed it.
We nonetheless added the following clarification in the discussion section: “Although we assumed an infinite time of death at the preferred temperature, one could add a parameter for longevity, representing the maximum time until death for the organism. Doing this might be useful to compare species with very different life expectancies but the increase in the complexity of the function and associated calculations prevented exploring that parameter in this study. Moreover, we focused on creating a TDC to identify restrictions on survival at stressful temperatures, rather than life expectancy at benign temperatures, which likely depends on many other factors”
- b) Second, the authors provide example data (lines 131-141) using U. ornatus.
Although they list a longevity for this animal (8760 hours), they do not actually
use this longevity in their calculations per their R code. I would like to know why
the longevity is provided in the manuscript example but not used in the R code.
We do not include this data anymore.
- c) Finally, if the Longevity variable is omitted, then the curve must be “flipped” about
the x-axis (i.e. a negative sign added to reverse the sign of all values). This is
reflected in the R code but not mentioned. Perhaps this is intuitive for those with
a deep understanding of mathematics, but a note in the R code about this orsomewhere may be helpful. All in all, I think the longevity variable needs more
consideration.
Also true, we corrected equation 1 and the discussion section, as commented above.
The minus sign simply adds the decremental effect of temperature in the time to death. This is now clarified in the script, too.
- There are many discrepancies (and some inaccuracies) between the manuscript and R scripts. The longevity value given in the example (U. ornatus) is incorrect. Three years equates to 26,280 hrs, not 8760 h as given by the authors. Moreover, some of the other values in this example do not match the R code (and the R scripts don’t match one another)..
Thank you for your patience and good sight.
We cleaned the reference to longevity, as we no longer use it. This reference was a remanent of an older version of the function. Although it makes sense to include longevity, doing it would require a set of evaluations that we will better address in the next article.
For example, one script lists U. ornatus CTMax as 47 and the other lists it as 45
This also comes from different versions along which we ended up deciding to choose different values. The newly submitted Rscript, titled: “DR and curve calculation,” does no longer use the incorrect value 47 and we corrected the values both in the manuscript and in the script.
“For PBT, VTmaxax, and CTmax, we used the median values of PBT= 35.5° [9], VTmax= 42.52°, and CTmax=46.3°C, respectively, from datasets “uro3” and “ctmaxuro” [23].”
We rebuilt the graphs accordingly, and, as expected, the small change does not change the conclusions from the interspecific comparison nor the comparisons among vulnerability layers produced for U. ornatus.
The manuscript lists the value of VTM as 42.77 but the scripts have values of 42.5.
There were several options available to summarize the VTM (now VTmax) actual value for VTmax is 42.52, as obtained in a previous study and referred above. We now corrected in the methods and the scripts and rebuilt the graphs based on this value, with no observable effect on our conclusions.
One script lists the value of DR as 3.01 and the other lists it as 3.4. Neither of those
values of DR is provided in the manuscript version of the example (but should be).
We now formally calculated DR values using an nls equation and added the resultant values to the results in the main text.
“The decay rates were 4.14 for U. ornatus, 135.6 for X. vigilis, 0.95 for S. occidentalis, and 2.05 for S. quoyi.”
Finally, one of the scripts has a variable called TPMAX. I don’t know what that is, but assume based on context clues it is preferred body temperature.
TPMAx it was just the way one of us referred to the VTM in one of the previous versions that got mixed. This is now corrected to VTmax, to conform solicitations of referees in other articles published in different journals.
Given the issues in number 3 above, combined with the fact that there are still comments (potentially unaddressed criticisms? I hope not) from the authors in the supplement, it appears this paper may have been hastily submitted without a careful review. Or perhaps an older version of the supplement was accidentally submitted? I hate to levy that accusation, but I’m not sure what else to think. I think the authors should carefully review the details prior to resubmission.
We reviewed these details more carefully in this occasion. We thank you for your patience.
The authors must provide a better explanation for how DR (decay rate) is calculated.
They say that curves were simply “fit by the eye”. What does that mean? Is DR simply
from a regression of survival by temperature values? Is it a slope? I think the authors
should provide an explicit example of how DR was calculated for U. ornatus so it is clear.
We added the following explanation to the original definition of DR: “The decay rate represents the thermal sensitivity of time until death. Namely, how much the life expectancy of individuals decays per increase in body temperature. This parameter can also be interpreted in the opposite direction: how much time until death increases as body temperatures decrease below the CTmax. The decay rate can be estimated by fitting the curve to data on observed times to death at different temperatures through non-linear least square models. We used the R base function “nls”.
This includes a formal way to calculate DR from observed times to death at different temperatures, using a non-linear least squares model.
This is particularly important given the emphasis the authors put on DR in the
discussion. Something like Figure 1 in this paper would be useful:
Andrews, R. M., & Schwarzkopf, L. (2012). Thermal performance of squamate embryos with respect to climate, adult life history, and phylogeny. Biological Journal of the Linnean Society, 106(4), 851-864.
We added the supplementary figure 3, showing how the variation in DR changes the shape of the curve.
I think the model needs to be cross validated with existing occurrence data for U. ornatus in some way. One concern I have with the TDC approach is that it might work somewhat like R^2values in simple regression - the more variables you add, the R^2 necessarily increases, although some of those variables may not actually be informative. Therefore, we would expect that the TDC would predict a much larger area of deleterious events than any of the other variables on their own. But is that actually useful? Put another way, under what circumstances would adding multiple measures of heat stress together result in fewer predicted deleterious events compared to just considering a single metric? As I look at the existing range of O.
ornatus, it appears that there are large areas of the range that your TDC model indicates should be thermally stressful and cause population declines. Yet, the lizards are there. Is there a way you can compare the distribution of deleterious events in Figure 2 with the existing occurrence data from GBIF, iNaturalist, something of that sort, to see which parameter best predicts thecurrent range? I think this would fully validate your TDC. At the very least, you should provide a range map of the species and discuss how this range map relates to the results shown in Figure 2.
As reviewer 2 may guess, although very relevant, those questions would be better addressed in a whole new study. Herein, we are simply showing that predictions of thermal vulnerability may change dramatically depending on the metric used and that there seem to be geographic biases arising from using less complete metrics of thermal tolerance.
How much thermal stress restricts the distribution of any given species is a different, although of course related, problem in the evaluation of climatic vulnerability.
We nonetheless followed the reviewer’s suggestions and added a simple comparison of the Urosaurus locations with the vulnerability layers, suggesting that Urosaurus mostly occurs where none of the layers predict thermal vulnerability, with some locations identified by the TDC and the VTmax as vulnerable.
Minor concerns
For equation 2, did you mean to use subscripts rather than commas? Also, why is there a comma at the very end of the equation (delta Tb,?). Finally, I assume that t= time and Tb = body temperature and Tb, t = body temperature at a particular time, but I don’t think this is explicitly stated. It should be.
We removed the commas and defined each parameter explicitly.
Line 158 - again, the PBT in the manuscript doesn’t match the R code. Has this influenced your predictions greatly
Actually not, we initially simplified from 35.9 to 35 because we needed to match that temperature with the .nc file dataset of number of hours per year spent at each temperature step, from the preferred body temperature to the CTmax, but that simplification is not needed anymore. We also corrected the value in the Rcode to 35.5 to match the literature [9]. Nor the output or the conclusions were visibly altered. Below 43 degrees, no deleterious events are detected by the TDTC.
Lines 166-169. Because this is the entire point of the paper (i.e. some new methods), you should give a brief example of how this was done. For example, I assume what you did is that, for each temperature from PBT to CTmax, you calculated the number of hours required for “damage”. Then you checked to see how many times that temperature was experienced for that amount of time at each geographic location. I think you should be explicit here.
We clarified the part in which we describe how many times the combination of temperature-time that leads to population damage happened. As written:
“To detect time-temperature combinations that should induce population damage, we extracted from the netCDF files: 1) the mean number of days where body temperatures exceeded CTmax (days/year), 2) the mean number of days where body temperature was above VTmax for at least four hours (days/year), 3) the percentage of years where body temperature was above preferred body temperatures (PBTs) for at least four hours over more than 60 days (index used in [9]). 4) The number of hours where body temperature exceeded lethal combinations of temperature and exposure time, as determined in the TDC (eq. 1), at a resolution of 0.2° incremental steps from the PBT to the CTmax (hours/year). All of these were yearly variables that were averaged across the 20 years period of the database”
Line 180 - wording is confusing here. Please edit.
We added a wider explanation:
Lines 180-181. Can you give the CTmax for each of the species so that the reader can compare CTmax with the curves? You could put it in the Figure next to the name of each species, or something like that. I think this is a major result of the paper and should be highlighted in the Results section in some way.
Sure. The results now read as follows:
“The fitted curves and published data show that species differ not only in their values of CTmax but also in their decay rates. CTmax of each species was 45.0°C for U. ornatus [23], 41.5°C for X. vigilis [36], 44.5°C for S. occidentalis [34], and 40.8°C for S. quoyi [32]. The decay rates were 4.14 for U. ornatus, 135.6 for X. vigilis, 0.95 for S. occidentalis, and 2.05 for S. quoyi. Figure 1 suggests that a species with a higher CTmax (e.g., S. occidentalis) may still tolerate a high temperature for less time than a species with similar CTmax would (e.g., X. vigilis).”
Lines 200-201. Please edit this text. The grammar needs to be more accurate.
We edited the grammar across the whole text.
Lines 233-234. No. You did not use longevity in your models per your R code. Why are you highlighting the use of longevity here?
See our previous response
Lines 243-244 - Do you mean the VTM “flags” an inflection point or that it IS an inflection point. Please clarify. I don’t understand what you mean that it “flags an inflection point”.
By “flags”, we mean “marks” or “signals” a change in the thermal sensitivity of time to death, we do not think it “is”, since this is just a model representing how the biological processes might occur. We also clarified the text more.
“The VTmax creates an inflection point in the thermal death curve, shifting it from convex to concave (i.e., slowing the decay rate at temperatures over the VTmax). This is supported by studies of U. ornatus showing that right below the VTmax (41°C), the time to death rises around 230h [21], while at the VTmax remains below 4h and at the CTmax happens below 1h [23]. This deceleration could be caused by the induction of protective physiological/biochemical processes, triggered when body temperatures rise, but studies need to ascertain whether the VTmax acts as such a trigger or these processes onset at lower or higher temperatures.”
Lines 256-260 - this is all the more reason why the calculation/estimation of DR should be more explicitly discussed in the methods.
Now they are. See our previous response.
Line 309 - what is TTDC?
The TDC, now corrected across the text.
Round 2
Reviewer 2 Report
I have evaluated the authors' responses to my original criticisms. I am mostly satisfied with the revisions and feel that the manuscript is much improved. I have no additional major criticisms of the work. There remain a few minor grammatical errors in the manuscript, which the authors can easily correct or could be corrected at the proofing stage. Some other relatively minor mistakes are listed below. I appreciate the work of the authors on the manuscript and the revisions and anticipate future work to further validate and utilize the TDC presented here.
Line 102 - you actually pose 3 questions, not 2
Line 207 - 46.3.0?
Line 228 - TVmax?
Line 451 - should "maximum age" be removed?
Author Response
I have evaluated the authors' responses to my original criticisms. I am mostly satisfied with the revisions and feel that the manuscript is much improved. I have no additional major criticisms of the work. There remain a few minor grammatical errors in the manuscript, which the authors can easily correct or could be corrected at the proofing stage. Some other relatively minor mistakes are listed below. I appreciate the work of the authors on the manuscript and the revisions and anticipate future work to further validate and utilize the TDC presented here.
Dear reviewer, thank you for your attentive review.
Line 102 - you actually pose 3 questions, not 2
Corrected.
Line 207 - 46.3.0?
Some automatic punctuation. Corrected
Line 228 - TVmax?
Line 451 - should "maximum age" be removed?
Removed.
We did a bit more proofing and changed a few more words. This was only to improve understanding/correction. The changes are marked.